# Identifying rates and risk factors for medication errors during hospitalization in the Australian Parkinson's disease population: A 3-year, multi-center study

Michael Bakker[1]*, Michaela E. Johnson [2], Lauren Corre[1], Deanna N. Mill [1,3], Xingzhuo Li[1], Richard J. Woodman[2], Jacinta L. Johnson[1,4]

1 SA Pharmacy, SA Health, Adelaide, South Australia, Australia, 2 College of Medicine and Public Health, Flinders University, Adelaide, South Australia, Australia, 3 School of Allied Health, University of Western Australia, Perth, Western Australia, Australia, 4 UniSA Clinical and Health Sciences, University of South Australia, Adelaide, South Australia, Australia

* Michael.bakker@sa.gov.au

**Data Availability Statement:** All relevant data are within the paper and its Supporting Information files.

## Abstract

### Background

Admission to hospital introduces risks for people with Parkinson's disease in maintaining continuity of their highly individualized medication regimens, which increases their risk of medication errors. This is of particular concern as omitted medications and irregular dosing can cause an immediate increase in an individual's symptoms as well as other adverse outcomes such as swallowing difficulties, aspiration pneumonia, frozen gait and even potentially fatal neuroleptic malignant type syndrome.

### Objective

To determine the occurrence and identify factors that contribute to Parkinson's medication errors in Australian hospitals.

### Methods

A retrospective discharge diagnosis code search identified all admissions for people with Parkinson's disease to three tertiary metropolitan hospitals in South Australia, Australia over a 3-year period. Of the 405 case notes reviewed 351 admissions met our inclusion criteria.

### Results

Medication prescribing (30.5%) and administration (85%) errors during admission were extremely common, with the most frequent errors related to administration of levodopa preparations (83%). A higher levodopa equivalent dosage, patients with a modified swallowing status or nil by mouth order during admission, and patients who did not have a pharmacist

**Funding:** D.N.M was supported by an Australian Government Research Training Program Scholarship at The University of Western Australia (https://www.dese.gov.au/research-block-grants/research-training-program). The Hospital Research Foundation provided financial support to cover publication costs (https://www.hospitalresearch.org.au). Neither of these funders had a role in study design, data collection and analysis, decision to publish, or preparation of the manuscript. This research did not receive any other specific grant from funding agencies in the public, commercial, or not-for-profit sectors.

**Competing interests:** The authors have declared that no competing interests exist.

led medication history within 24 hours of admission had significantly higher rates of medication errors.

## Conclusions

This study identified 3 major independent factors that increased the risk of errors during medication management for people with Parkinson's disease during hospitalization. Thus, targeting these areas for preventative interventions have the greatest chance of producing a clinically meaningful impact on the number of hospital medication errors occurring in the Parkinson's population.

## Introduction

Optimal and continued medication management is critical for maintaining the quality of life of people with Parkinson's disease. If Parkinson's medications are not given, patients may be unable to swallow causing a high risk of aspiration, they may be unable to speak or move and will likely become more dependent on others for assistance [1–4]. Continuity of treatment is important, with irregular dosing increasing the chance for frozen gait [5] that in turn increases the risk of falls and fractures. At worst abrupt withdrawal of dopaminergic medications can lead to the development of potentially fatal neuroleptic malignant type syndrome [2,6,7]. Patients taking medications to manage Parkinson's disease are susceptible to drug-food [8,9], drug-drug and drug-disease interactions [10], thus even if medication administration continues as normal; changes to timing of administration and introducing new medications can also prove detrimental, highlighting the importance of vigilant medication monitoring [2,7,9,11].

As Parkinson's disease progresses the pharmacological regimen required to manage the motor and non-motor symptoms becomes increasingly complex due to reduced responsiveness to treatment and the progressive nature of symptoms as the disease advances [12]. Transferring to and from a health care facility presents a particularly hazardous time for maintaining continuity of information and medication supply to manage the disease [13]. If a patient's Parkinson's medications are not appropriately managed throughout the admission, the patient may experience discomfort, delays in recovery, longer length of stay [14] and worse overall outcomes [11,15,16].

Despite the potentially severe consequences of mismanaged Parkinson's disease medication during hospital admissions few studies have evaluated the frequency and factors that contribute to this issue. Of those studies available, most are based on a single center and/or have small sample sizes [16–22], with the exception of Lertxundi et. al., [23] that reported medication errors in people with Parkinson's' disease in Spain were not only associated with increased duration of their hospital stays but also with higher mortality rates. In Australia the impact of 'get it on time' campaigns have only resulted in a small number of audits and our recent study investigating the reporting of medication errors within the Parkinson's disease population in a single hospital in South Australia [24]. This study revealed that voluntary incident reporting systems dramatically under represent the true amount of medication errors that occur for people with Parkinson's disease while in hospital [24]. This highlights the need for a larger study to accurately demonstrate the frequency and factors that contribute to this problem to support the development and implementation of appropriate preventative strategies/interventions. Thus, the primary aims of this study were to identify the true rates and types of medication errors related to the treatment of Parkinson's disease during hospitalization in multiple

centers. The secondary aim of this study was to identify contributing factors that impact the rate of Parkinson's medication errors so that future preventative measures can be successfully implemented.

## Methods

### Ethics

This study was approved by the Central Adelaide Local Health Network Human Research Ethics Committee as a multisite study (HREC/18/CALHN/330). Consent was not required from participants as the research met the criteria for waiving consent according to the National Statement, section 2.3.10.

### Study design

To identify all admissions of people with Parkinson's disease to three tertiary metropolitan hospitals over 3 years (1/7/2013 to 30/9/2016), we assessed a total of 1,161 hospital beds by performing a retrospective discharge diagnosis code search. Admissions were included if the International Classification of Diseases 10th edition code (ICD-10) [25] for Parkinson's disease (G20) was present within the first five coded separation diagnoses.

### Study site

All hospitals included operate under the same general workflows underpinned by the same policies and procedures. Doctors at all three hospitals use the paper-based Australian standard 'National Inpatient Medication Chart' to record the patient's medications required during the admission. None of the hospitals has implemented ward based automated dispensing cabinet for medication administration. Thus, all three hospitals had essentially the same processes relating to charting, recording, storage and supply of medicines.

### Study population

Admissions were excluded if the admissions corresponding to the patient had privacy restrictions, or if the patient was not prescribed any medications for the treatment of their Parkinson's disease prior to or during hospitalization. No clinical information relating to cognitive state or disease stage could be collated as assessments such as the Hoehn and Yahr scale were not performed if the person was admitted to the hospital for a reason unrelated to their Parkinson's disease e.g., surgery.

### Data collection

Investigators retrospectively reviewed full medical records for admission/s identified in the ICD-10 search and extracted data using a standardized data collection form (individual patient data can be viewed in S1 Table). Records of patients seen by a pharmacist during their hospital stay included a pharmacist led medication history document in addition to their 'National Inpatient Medication Chart'. Data collected included patient demographics, admission details, pharmacological management of the patient's Parkinson's disease prior to and during admission. Using a recognized conversion tool [26], patient's medication regimens were converted to single figures called 'Levodopa Equivalent Dosage' (LED), which facilitates comparisons between different mediation regimens. We also recorded pharmacist input in the form of admission medication history taking, clinical pharmacy review, including pharmacist recommendations for management of patients that were nil by mouth, had swallowing difficulties and/or required crushed medications. as well discharge medication reconciliation,

## Operational definitions

Medication errors were recorded and classified into two phases- the prescription phase, when medications were written on the patients' chart at admission, and the administration phase, when the medication was given to the patient. Prescribing errors on admission were classified as: omission error, incorrect dose, incorrect strength, incorrect medication, incorrect timing (+/- 30 minutes of the charted time) or incorrect formulation. Administration errors identified during their hospital stay were classified as dosed early, delayed dosing or omission error. Omission errors were identified by the administering nurses' annotations on the patients' chart to the effect of 'N/A', not given or absence of a signature to verify the dose was given at the appropriate time. Incorrect timing errors e.g., dosed early and delayed dose, were identified by assessing the time that a medication was administered according to the patients' chart. Incorrect medication errors (medication, dose, strength and formulation) could be identified by reviewing patient charts for nursing staff annotations. For example, an annotation that a patient was administered the slow-release levodopa instead of their standard release medication.

Where the reason why a certain medication was withheld/omitted was documented on the patient's chart this was recorded. No investigation was conducted regarding the clinical appropriateness of this decision e.g., no validation of whether it was appropriate or correct for that specific patient. The number of times an inappropriate medicine was administered was also recorded. An inappropriate medicine was described for the purpose of this study as a medicine that was pharmacologically inappropriate, opposing the effects of dopamine [27]. Agents included were antipsychotics, metoclopramide, prochlorperazine, bupropion, phenytoin and lithium. Low dose quetiapine (25 mg or less) was not included as an inappropriate medicine, as locally it is the accepted treatment for behavioral management in Parkinson's disease.

## Statistical analysis

Descriptive statistics were utilized to summarize patient characteristics, admission characteristics and outcomes investigated across the entire cohort and pre-defined subgroups (patients with modified swallowing status or nil by mouth order, and presence of a pharmacist led medication history).

Multiple imputation using predictive mean matching was used to account for missing data.

The association between Parkinson's medication related incident rate and dosage form modification, pharmacist led medication history, and LED were examined using negative binomial regression. Covariates age and pre-admission living arrangement were also included in the model. Hosmer-Lemeshaw test statistic was calculated to assess for model fit. Interactions between dosage form modification, pharmacist led medication history, and pre-admission living arrangement were also tested. Sensitivity analyses were performed by exclusion of significant outlying observations. Outlying observations were identified using conventional definition of more than 1.5 times interquartile range above the first quartile or below the first quartile. However, most extreme outlying observations were included in the sensitivity analyses as the data were found to be accurate for those people following further assessment of their case notes.

As the opportunity for errors increases with length of stay, analysis of total number of errors against length of stay is subject to innate confounding. Thus, we examined the relationship between the number of Parkinson's medication errors within the first 48 hours of admission and total length of stay using zero-truncated negative binomial regression. Other covariates included in the model were dosage form modification, pharmacist led medication history,

LED, pre-admission living arrangement and age. This analysis was also restricted to admissions longer than 48 hours.

All analyses were performed in Stata version 14 (StataCorp). Statistical significance was considered for p <0.05 (2-sided).

## Results

A total of 405 hospital admissions for people with Parkinson's disease were examined, with 351 (87%) patients included and 54 (13%) patients excluded for meeting and failing to meet our inclusion criteria, respectively. Patient demographics are summarized in Table 1. In brief, the patients included were predominately over 78 years old, male (62%) and had hospital stay durations of 6 days or longer. Increasing age decreased the frequency of medication errors, with a 14% reduction in risk for every ten-year increase in age (IRR = 0.86, p = 0.025, 95% CI 0.76–0.98). Conversely, residing in a nursing home immediately prior to hospital admission increased the risk of medication errors two-fold (IRR = 2.15, p<0.001, 95% CI 1.55–2.98).

Over 30% of all medications involved an incident during the prescribing phase of the hospital admission, with the most common type of incident being incorrect timing (59.5%), followed by omission error (15%) and incorrect dosage (14.5%). Table 2 further describes the types of prescribing medication errors made on patient charts at the time of hospital admission. A total of 85% of all case notes reviewed contained one or more Parkinson's medication errors during the administration phase. Omission error was responsible for the largest proportion (55.9%) of administration errors (Table 3). Levodopa-based treatments accounted for 83% of all administration errors, reflecting the commonality of levodopa as the main therapy used in Parkinson's disease. The frequency of administration errors for levodopa preparations as well as all other medications are detailed in Table 4. Surprisingly, of the 2,013 identified administration errors only 42% had a reason documented on the patients' chart. Note that we did not validate the clinical appropriateness for the stated reasons for the dosed early, delayed dose or omission error as this was outside of the scope of our study. Error rates were similar across the three tertiary hospitals included in this study.

Other factors we observed to impact the rate of medication errors included absence of a pharmacist led medication history, or if it was completed more than 24 hours after admission which occurred in 24% and 43% of people, respectively; together this absence/delay resulted in individuals being 50% more likely to have a Parkinson's medication error (IRR = 1.57, p = 0.010, 95% CI 1.12–2.21). Our multivariate negative binomial regression also revealed for

**Table 1. Patient characteristics for the people with Parkinson's disease included in this study.**

| Patient Characteristics | Study Group (*n* = 351) |
|---|---|
| Age in years, median (range) | 79 (36–98) |
| Male, *n* (%) | 1. (62) |
| Length of stay in days, median (range) | 6 (1–71) |
| Previous residence listed as Nursing Home, *n* (%) | 104 (30) |
| Pharmacist led medication history completed | |
| Yes, *n* (%) | 266 (76) |
| Completed more than 24 hours after admission, *n* (%) | 152 (43) |
| LED[a] of treatment regimen in mg, median (range) | 663 mg (75–3800) |
| Modified swallow status or nil by mouth during admission, *n* (%) | 101 (29) |

[a]LED = Levodopa Equivalent Dosage.

**Table 2. Types of Parkinson's medication errors that occurred when prescribing on first medication chart following admission for people with Parkinson's disease.**

| Type of incident | Prescribing errors | |
|---|---|---|
| | *n* | % |
| Incorrect dosage | 25 | 14.5 |
| Incorrect timing | 103 | 59.5 |
| Omission error | 26 | 15 |
| Wrong medication | 4 | 2.3 |
| Wrong formulation | 8 | 4.6 |
| Wrong strength | 7 | 4 |

Total number of medications prescribed was *n* = 568, of those *n* = 173 had prescribing errors.

every 200 units higher LED, patients were 14% more likely to have a Parkinson's medication error during their hospital stay (IRR = 1.14, p<0.001, 95% CI 1.06–1.22). We found 29% of admissions required alterations to medication formulation due to difficulty swallowing solid dose oral forms, and that this subgroup had a 38% increased risk of errors (IRR = 1.38, p = 0.023, 95% CI 1.05–1.81). These patients' medication errors were often clustered together as opposed to being spread out over the course of the admission. We also found that this group of patients were more likely to have an increased length of hospital stay (IRR = 1.45, p = 0.002, 95% CI 1.15–1.85). More generally, we found no association between number of medication errors within the first 48 hours of admission and total length of stay for people with Parkinson's disease.

Inappropriate medicines were prescribed to 15% of patients, with over half of these patients administered at least one dose. Overall, 140 doses of inappropriate medications were administered, including 35 doses of metoclopramide, 33 doses of risperidone, 6 doses of haloperidol and 66 doses of other medications such as buspirone, olanzapine or prochlorperazine.

## Discussion

We identified the rates and types of errors that people with Parkinson's disease face in their medication management at admission and during their stay at three hospitals across South Australia. We found a staggering 85% of patients had at least 1 medication administration incident during admission. This is the first multi-center study focusing on this issue in Australia, with our results reflecting medication errors reported by Lertxundi et. al., [23] in the Spanish Parkinson's' disease population. Our results are also consistent with other smaller and/or single center studies conducted in the UK, USA, China and New Zealand [16–22]. Taken together, these studies indicate that people with Parkinson's disease are particularly vulnerable

**Table 3. Types of Parkinson's medication errors that occurred during the administration phase for people with Parkinson's disease during admission.**

| Type of incident | Administration errors | |
|---|---|---|
| | *n* | % |
| Omission error | 1125 | 55.9 |
| Dosed early | 93 | 4.6 |
| Delayed dose | 763 | 37.9 |
| Wrong order | 32 | 1.6 |

Total number of administration errors *n* = 2,013.

**Table 4. Frequency of medication administration errors during admission, listed by medication.**

| Medication | Errors identified | |
|---|---|---|
| | *n* | % |
| Amantidine | 15 | 0.7 |
| Apomorphine | 2 | 0.1 |
| Cabergoline | 6 | 0.3 |
| Entacapone | 133 | 6.6 |
| Levodopa/benserazide | 295 | 14.7 |
| Levodopa/benserazide SR[a] | 19 | 0.9 |
| Levodopa/carbidopa | 1006 | 50 |
| Levodopa/carbidopa SR[a] | 55 | 2.7 |
| Levodopa/carbidopa/entacapone | 301 | 15 |
| Pramipexole | 68 | 3.4 |
| Pramipexole ER[b] | 48 | 2.4 |
| Rasagiline | 38 | 1.9 |
| Rotigotine | 27 | 1.3 |

[a]SR = Slow release.

[b]ER = Extended release.

Total number of administration errors *n* = 2,013.

to medication errors while in hospital, and that this issue occurs internationally. Thus, highlighting the urgent need to identify factors that contribute to this problem to ultimately improve patient outcomes for people with Parkinson's disease.

Within the Parkinson's disease population, patients with a modified swallowing status or nil by mouth order were identified as 'higher risk', as they encountered significantly more medication errors and had an increased risk for longer length of hospital stay. If patients were unable to swallow their regular solid dose medications, they likely required an alternative medication formulation, this transition may have been responsible for the observed increased rate of medication errors for this subgroup. Published data flag surgical and post-operative care as acute risks to medication management for people with Parkinson's disease [28–33]. Based on our results we suggest people with Parkinson's disease who have swallowing difficulties should be added to this acute risk list so that this subgroup can receive extra care and consistent management of medications during hospital stays to lower the rate of errors. In support of this, Derry et. al., [29] found that 'unable to swallow' was recorded as the reason for 14% of their observed missed or omitted doses in people with Parkinson's disease during surgical admissions. Developing a local protocol to inform treatment alterations necessary to maintain continuity of levodopa therapy when a person with Parkinson's disease is unable to swallow their medication e.g. switching to transdermal rotigotine, could help to circumvent the increased risk for this specific subgroup with the Parkinson's disease population.

As expected, the greater the complexity of a patients Parkinson's treatment, expressed as a higher LED, the greater the risk of medication errors. Based on the broad range of LED documented in this study, we can see that when patients commence treatment they have an established level of meaningful risk of errors and that this develops to a very high risk of errors as their LED increases over the course of disease. One approach to help people with Parkinson's disease have more control over their medication management during hospital admissions is to have a 'go bag' of their current medications that is immediately ready to be taken to hospital in the event of an emergency. Since our earlier study demonstrating that medication errors are

underreported in people with Parkinson's disease in Australia [24], this 'go bag' initiative has been disseminated to the community by Parkinson's South Australia, a nonprofit organization that provides support and information to people living with Parkinson's disease.

Many other solutions or initiatives to improve medication management in hospitals require the development of new resources or specialized training programs for staff [34,35]. In this study we found that in the event a pharmacist was unable to undertake a medication history for the patient within the first 24 hours of admission these patients had a 1.5-fold increased risk of Parkinson's medication errors. When we compared a pharmacist led medication history to the first chart medication completed by other health professionals, we found that just over 30% of Parkinson's medications were charted incorrectly. We believe these data indicate that people with Parkinson's disease should be ranked as of 'high importance' when developing clinical prioritization tools for pharmacists. Pharmacists supporting prescribers by completing a thorough medication history when a person with Parkinson's disease is first admitted to the hospital offers a realistic solution that can be easily implemented to reduce error rates, as clinical pharmacists are often already present in large hospital pharmacy departments. This recommendation aligns with suggestions for people with Parkinson's disease previously published by the Institute for safe medication practices [3]. In contrast to our results, a recent study by Cowley et. al., [36] assessing 84 patients in a single center found no correlation between administration-related medication errors and the time taken to complete a medical history by a pharmacist/pharmacy technician. However, the latter study excluded patients who had other health professionals record their medication histories therefore, we cannot determine if those patients without a pharmacy led medication history had a greater rate of errors, like that observed in our study. The discrepancy between our study and Cowley et. al., [36] on the influence of the timing of a pharmacist led medication history can have on error rates could be due to our study including prescribing of inappropriate medications in addition to omission errors and incorrect timing of medication under 'medication errors', thus perhaps it is inappropriate medications that drives the association found in this study.

While hospitalized 15% of patients were prescribed an inappropriate medicine, with more than half of these patients administered at least one dose. No consistent approach to discourage prescribing inappropriate medications was observed. For example, some individual pharmacists used the adverse drug reactions and intolerances section of the 'National Inpatient Medication Chart' to highlight antidopaminergic drugs. However, this was only a very limited number of clinicians, and it appears there may have been instances where this documentation was present and inappropriate medication was still charted. A limitation of the nature of a retrospective audit is that we are unable to determine if the advice indicating an inappropriate medication was recorded on the chart was present at the time that the medication was charted or whether it was subsequently added after the error had occurred to prevent further medication errors. Published data in this area mirrors our findings of inappropriate prescribing and highlights the presence of anti-dopaminergic medications in increasing fall rates, morbidity and mortality during hospitalization [17,37–39]. Initiatives such as using the adverse drug reaction box to highlight medications to avoid need to be promoted within the healthcare setting. Other initiatives that have a proven benefit in reducing medication related harm and errors in the general patient cohort [40–42] such as electronic health records, electronic prescribing, pharmacist involvement in accident and emergency, and partnered-pharmacist charting of medicines could also reduce these preventable errors in this vulnerable patient group [43].

A limitation of this study was that it was conducted in hospitals with paper-based records which makes it difficult to establish the sequence of events, inaccurate recording of information on charts/ progress notes and to evaluate the relationship between medication errors and

the clinical impact to a patient. Repeating this study in the presence of electronic health records and wards with automated dispensing cabinets would improve the precision and accuracy of the data and reduce gaps in the information that we are able to pull from patient records. Another limitation was that the retrospective nature of the study meant the factors evaluated were based on what information could be collected from all patient records. For example, if the person was admitted to the hospital for a reason unrelated to their Parkinson's disease, e.g., surgery, assessment of their disease stage was not performed, thus this prevented us from analyzing associations between disease stage and medication errors. A major strength of our study is the large sample size and that it was conducted across multiple hospital sites. From here future research could explore the root causes for common error types and interventions could be developed using corresponding evidence-based behavior change techniques.

## Conclusion

In summary, this study assessed three aspects of medication management for people with Parkinson's disease during hospitalization: inaccurate prescribing of medicines, medication administration errors, and prescribing of inappropriate medicines. We identified that more complex medication regimens and being unable to swallow solid dose oral medicines were significant risks for medication errors and that pharmacist led medication histories were able to reduce the frequency of medication errors. We recommend adjusting health system approaches to target these particular factors, and suggest working with people with Parkinson's disease to utilize existing levers such as staff training, guideline development and clinical prioritization of people with Parkinson's disease for pharmacist review when they first present to hospital. These strategies may represent achievable solutions to reduce preventable medication errors in this patient group.

## Supporting information

**S1 Table. Raw data collected for each hospital admission included in this study's statistical analysis (*n = 351*).**
(XLSX)

## Acknowledgments

The authors would like to thank pharmacy students Lauren Rask-Nielsen, Sophie Benger, Tuyet Linh Sophia Quach, Truc Ngoc Thi Bui and Vanessa Koumi for their assistance with data collection and input throughout the study.

## Author Contributions

**Conceptualization:** Michael Bakker, Lauren Corre, Deanna N. Mill, Jacinta L. Johnson.

**Data curation:** Lauren Corre, Deanna N. Mill, Richard J. Woodman.

**Formal analysis:** Michael Bakker, Xingzhuo Li.

**Supervision:** Jacinta L. Johnson.

**Writing – original draft:** Michael Bakker.

**Writing – review & editing:** Michaela E. Johnson, Lauren Corre, Deanna N. Mill, Jacinta L. Johnson.

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
