## [Decision Letter · Decision Letter 0]

8 Mar 2022

PONE-D-21-35261Identifying rates and risk factors for medication incidents during hospitalization in the Australian Parkinson’s disease population: A 3-year, multi-center studyPLOS ONE

Dear Dr. Johnson,

Thank you for submitting your manuscript to PLOS ONE. After careful consideration, we feel that it has merit but does not fully meet PLOS ONE’s publication criteria as it currently stands. Therefore, we invite you to submit a revised version of the manuscript that addresses the points raised during the review process.

We look forward to receiving your revised manuscript.

Kind regards,

Ismaeel Yunusa, PharmD, PhD

Academic Editor

PLOS ONE

Journal Requirements:

Additional Editor Comments :

Please ensure you follow the EQUATOR/STROBE recommendation in reporting the revised manuscript. 

Reviewers' comments:

Reviewer's Responses to Questions

**Comments to the Author**

1. Is the manuscript technically sound, and do the data support the conclusions?

Reviewer #1: Yes

Reviewer #2: Yes

2. Has the statistical analysis been performed appropriately and rigorously? 

Reviewer #1: I Don't Know

Reviewer #2: Yes

3. Have the authors made all data underlying the findings in their manuscript fully available?

Reviewer #1: Yes

Reviewer #2: Yes

4. Is the manuscript presented in an intelligible fashion and written in standard English?

Reviewer #1: No

Reviewer #2: Yes

5. Review Comments to the Author

Reviewer #1: Congratulations on the study! It has contributed some very important data and insights to support the optimisation of medication management in patients with Parkinson's disease. It would be interested to see other sites in Australia, especially digital hospitals, to conduct similar studies.

General Feedback:

- Be consistent with the use of various terms throughout the article: Medications vs medicines vs drugs; Parkinson’s disease vs Parkinson’s; Parkinson’s disease medication vs Parkinson’s medication; Automatic dispensing cabinets (Line 102) vs automated pharmacy distribution system (Line 302); Prescribing incident (Line 122) vs prescribing medication errors (179)

- Consider breaking down some of the long sentences into several shorter ones. It would make the article easier to read and understand.

Specific Feedback:

- Line 71 “… have low sample sizes” – consider changing to ‘low’ to ‘small’; a bit odd to use ‘low’ to describe sample sizes

- Line 75 “… our recent study” – what is the reference to that study?

- Line 80 “… so solutions can be sort” – consider re-wording to something like ‘… to support the development and implementation of appropriate preventative strategies/interventions’

- Line 100 – 102 “They use paper-based recording… automated dispensing cabinets” – consider breaking down this sentence into several shorter sentence. It is difficult to comprehend in its current form. For example, “Doctors at all three hospitals use the paper-based Australian National Inpatient Medication Chart to record the patient’s medications required during the admission. None of the hospitals has implemented ward based automated dispensing cabinet for medication administration by nurses.”

- Line 114 – 115 “…which was converted to a single figure…” – consider making this a separate sentence and provide a description of LED.

- Line 116 “… input on discharge” – this is unclear. Are you referring to discharge medication reconciliation +/- number of pharmacy interventions?

- Line 116 consider outlining the pharmacy input in the order of the patient’s journey, i.e. admission medication history taking, clinical pharmacy review, discharge reconciliation. Utilise standard clinical pharmacy terminologies.

- Line 117 Is “the management of patients…” referring to management by pharmacist only or any clinicians?

- Line 122 – replace ‘omitted drugs’ with ‘omission error’ which is a standard term used in medication safety literature

- Line 123 replace ‘drug’ with ‘medication’

- Line 125 replace ‘omitted drugs’ with ‘omission’

- Line 125 – 127 this sentence is unclear and confusing. I needed to read it a few times to understand it. When it says “if a reason for withholding a [medication] was noted on the chart”, does it mean the NIMC or the patient’s chart (progress notes)? Did the investigator record the reasons for withholding (W) as per NIMC or they would find out the actual reasons for (W) on the NIMC?

- Line 143 the word ‘led’ is missing from “… pharmacist medication history”.

- Line 153 replace ‘vs’ with ‘against’

- Line 157 replace ‘levodopa equivalent dose’ with the abbreviation ‘LED’

- Table 1 – what is the mean/median and range for the number of Parkinson’s medications for the study population?

- Line 177 “Over a third of all medications involved…” – what is the total number (n) of medications involved? It is noted in the abstract that 31% of cases had prescribing errors (= 109 patients). If this sentence is referring to the 31% mentioned there, please note that 31% does not equal to over a third.

- Line 182 replace ‘dose omitted’ with ‘omission error’

- Line 187 “only 42% had a reason documented on the chart…” – please clarify this statement. Do you mean nurses actually document a reason for omission, dosed early, or delayed dose on the medication chart? Or in the patient’s notes? Or are you referring to the reason for withholding a dose of Parkinson’s medication? If so, what is the total number of medications or doses that the 42% is referring to? Also refer to comment for Line 125 – 127.

- Table 2 replace ‘drug omitted’ to ‘omission error’ and ‘drug’ with ‘medication’

- Table 3 replace ‘drug omitted’ to ‘omission error’

- Line 222 – 223 “No consistent approach to discourage prescribing inappropriate medicines were observed” – this sentence would be considered more for the Discussion section. Consider describing existing high risk alert strategies observed at the study sites and discuss what kind of interventions should be considered e.g., alert sticker, clinical decision support.

- Line 237 consider expanding more on the reason why alterations to medication formulation would contribute to high risk?

- Line 267 use lowercase ‘m’ for ‘medications’

- Line 293 “… such practice need to become widespread” – this expression is a bit odd. Consider re-wording to something like “… such practices need to be promoted/shared/standardised within the healthcare setting” or something similar.

Reviewer #2: very good study I recommend publication after minor revision

I have few comments that might improve the manuscript:

1-The methodology section can be presented better if have sub-headings on ethics, study design, study site, study population (inc/exc criteria), data collection, operational definitions, and statistics.

2-The abbreviations in the tables must be defined in the footnotes

3-Authors have used the term medication incidents, which is not very common in the pharmacy field. What if the author considers using the term "Medication errors or medication-related problems"?

4-Authors have evaluated the factors leading to medication incidents and in the methodology section, they have described those reasons of incidents were noted if they were presented in the medication chart. I have one confusion here if the pharmacist has visited the patient within 24 hours can be identified from the record, can the authors describe how other factors were identified? were they being routinely noted in the patients` records?

5-The conclusion section is broad, please be specific in conclusion with a separate heading.

6-This paper lacks recommendations for pharmacists, healthcare authorities, and physicians, as well as for future research.

7-This study is accompanied by a few limitations as well as strengths, but the authors did not provide them at the end of the discussion section. For example, researcher bias is common in this study, the evaluation of factors all depends on the researcher`s ability to assess them. Strength is sample size from three large hospitals

6. PLOS authors have the option to publish the peer review history of their article (what does this mean?). If published, this will include your full peer review and any attached files.

Reviewer #1: No

Reviewer #2: **Yes: **Abdullah S Alanazi

---

## [Author Response · Author response to Decision Letter 0]

27 Mar 2022

Response to journal and reviewer comments

We appreciate the time of the journal staff and reviewers and thank them for providing thoughtful and constructive feedback to improve our manuscript (PONE-D-21-35261). We have incorporated the suggested changes or responded to comments as outlined below and as indicated by tracked changes in the revised manuscript. 

Journal Requirements:

We have ensured the submission meets PLOS ONE’s style requirements, including naming files for the title page and main body as per the guidelines. 

We have now included our raw data as a supporting Table so that readers can access all data collected for the analysis conducted in the presented study. This table is referred to in the methods section (line 116-117) and at the end of the manuscript under the ‘supporting information’ heading (line 466-468). 

We have removed the phrasing ‘data not shown’ from the manuscript as requested, all data are now presented directly in the paper and/or available in the supplementary file, which is described in our response to comment 2 above.

We have reviewed the reference list and confirm it is complete and correct. No retracted references have been cited. 

Additional Editor Comments :

Please ensure you follow the EQUATOR/STROBE recommendation in reporting the revised manuscript. 

We have ensured that the manuscript follows STROBE recommendations.

Reviewer #1

Congratulations on the study! It has contributed some very important data and insights to support the optimisation of medication management in patients with Parkinson's disease. It would be interested to see other sites in Australia, especially digital hospitals, to conduct similar studies.

Thank you! We appreciate your interest in and enthusiasm towards this work.

General Feedback:

- Be consistent with the use of various terms throughout the article: Medications vs medicines vs drugs; Parkinson’s disease vs Parkinson’s; Parkinson’s disease medication vs Parkinson’s medication; Automatic dispensing cabinets (Line 102) vs automated pharmacy distribution system (Line 302); Prescribing incident (Line 122) vs prescribing medication errors (179)

We have reviewed the manuscript and amended the wording so that phrases are consistent throughout. Please see tracked changes. We have ensured consistency in the phrases highlighted above by the reviewer, and amended any other inconsistencies identified, for example, we use the term ‘automated dispensing cabinets’ both on line 103 and line 323-324. 

- Consider breaking down some of the long sentences into several shorter ones. It would make the article easier to read and understand.

We have reviewed the manuscript and instances where long sentences were observed we have split these into multiple shorter sentences to help improve readability. 

Specific Feedback:

- Line 71 “… have low sample sizes” – consider changing to ‘low’ to ‘small’; a bit odd to use ‘low’ to describe sample sizes

We have changed ‘low’ to ‘small’ sample sizes on line 70.

- Line 75 “… our recent study” – what is the reference to that study?

Thank you for identifying our missing reference. We have rectified this by including the reference to our recent study by Mill et al 2020 at the end of the sentence on line 75. 

- Line 80 “… so solutions can be sort” – consider re-wording to something like ‘… to support the development and implementation of appropriate preventative strategies/interventions’

We have changed the highlighted text to incorporate reviewer 1’s suggested phrasing at the end of the sentence on line 79-80. 

- Line 100 – 102 “They use paper-based recording… automated dispensing cabinets” – consider breaking down this sentence into several shorter sentence. It is difficult to comprehend in its current form. For example, “Doctors at all three hospitals use the paper-based Australian National Inpatient Medication Chart to record the patient’s medications required during the admission. None of the hospitals has implemented ward based automated dispensing cabinet for medication administration by nurses.” 

We have rephrased the long sentence into two shorter sentences as recommended by reviewer 1 to improve clarity, line 101-104. 

- Line 114 – 115 “…which was converted to a single figure…” – consider making this a separate sentence and provide a description of LED.

We have split the description of LED onto its own line as suggested. This can be seen in the ‘data collection’ section of the methods, line 120-122. 

- Line 116 “… input on discharge” – this is unclear. Are you referring to discharge medication reconciliation +/- number of pharmacy interventions?

Yes, we were referring to a pharmacist checking which medications a person has been prescribed when they leave the hospital. We have changed the phrasing in the indicated sentence to ‘discharge medication reconciliation’ as per reviewer 1’s suggestion, line 125-126. 

- Line 116 consider outlining the pharmacy input in the order of the patient’s journey, i.e. admission medication history taking, clinical pharmacy review, discharge reconciliation. Utilise standard clinical pharmacy terminologies.

We have restructured the order of the sentence on line 122-126 to follow a chronological flow of the pharmacist role from patient admission to discharge, specifically using the standard clinical pharmacy terminologies outlined by reviewer 1.

- Line 117 Is “the management of patients…” referring to management by pharmacist only or any clinicians? 

The ‘management of patients’ referred to pharmacist helping by advising clinicians on which medications were most appropriate for those patients with a modification swallowing status. We have modified the phrasing of this sentence to clarify. It now reads “….including pharmacist recommendations for management of patients that were nil by mouth…” , now line 124.

- Line 122 – replace ‘omitted drugs’ with ‘omission error’ which is a standard term used in medication safety literature

We have changed ‘omitted drugs’ to ‘omission error’ as seen on line 131-132. 

- Line 123 replace ‘drug’ with ‘medication’

The sentence spanning line 131-133 regarding the classification of prescribing incidents has had ‘drug’ replaced with ‘medication’.

- Line 125 replace ‘omitted drugs’ with ‘omission’

We have changed ‘omitted drugs’ to ‘omission’ as seen on line 134. 

- Line 125 – 127 this sentence is unclear and confusing. I needed to read it a few times to understand it. When it says “if a reason for withholding a [medication] was noted on the chart”, does it mean the NIMC or the patient’s chart (progress notes)? Did the investigator record the reasons for withholding (W) as per NIMC or they would find out the actual reasons for (W) on the NIMC?

We have split the indicated sentence into two sentence and rephrased to improve the clarity. The new sentences for this topic are now as follows “Where the reason why a certain medication was withheld/omitted was documented on the patient’s chart this was recorded. No investigation was conducted regarding the clinical appropriateness of this decision e.g., no validation of whether it was appropriate or correct for that specific patient”, line 143-145. 

- Line 143 the word ‘led’ is missing from “… pharmacist medication history”.

Thank you for identifying this typographical error, ‘led’ has now been added to the highlighted sentence on line 160.

- Line 153 replace ‘vs’ with ‘against’

This change has been made as suggested on line 170.

- Line 157 replace ‘levodopa equivalent dose’ with the abbreviation ‘LED’

The abbreviation ‘LED’ has replaced ‘levodopa equivalent dose’ on line 174.

- Table 1 – what is the mean/median and range for the number of Parkinson’s medications for the study population?

In line with literature in this field we have chosen to represent patients Parkinson’s medication by LED rather than the number of Parkinson’s medication as the former is more reflective of the complexity of their medication regime and allows easier comparison between patients. The median and range for LED of the patient population can be viewed in Table 1 on page 7/8. 

- Line 177 “Over a third of all medications involved…” – what is the total number (n) of medications involved? It is noted in the abstract that 31% of cases had prescribing errors (= 109 patients). If this sentence is referring to the 31% mentioned there, please note that 31% does not equal to over a third.

The total number of medications prescribed (n=568) has been include as part of Table 2’s description on line 208. We thank the reviewer for noting this oversight and have now corrected the indicated sentence to read “Over 30% of all medications…” on line 192.

- Line 182 replace ‘dose omitted’ with ‘omission error’

We have replaced ‘dose omitted’ with ‘omission error’ on line 197.

- Line 187 “only 42% had a reason documented on the chart…” – please clarify this statement. Do you mean nurses actually document a reason for omission, dosed early, or delayed dose on the medication chart? Or in the patient’s notes? Or are you referring to the reason for withholding a dose of Parkinson’s medication? If so, what is the total number of medications or doses that the 42% is referring to? Also refer to comment for Line 125 – 127.

Yes, in some instances the reasons for omissions, delayed or early dosing were indeed documented on the patients’ charts by nursing or pharmacy staff using short codes. The 42% statistic relates to reasons documented on the chart for any administration error (e.g., combined for dosed early, delayed dose or omission error), not just reasons for withholding medications. We have amended the text to clarify this. These changes can be viewed on line 201-204.

- Table 2 replace ‘drug omitted’ to ‘omission error’ and ‘drug’ with ‘medication’

We have replaced ‘dose omitted’ with ‘omission error’ and ‘drug’ with ‘medication’ in Table 2, seen on page 8/9.

- Table 3 replace ‘drug omitted’ to ‘omission error’

We have replaced ‘dose omitted’ with ‘omission error’ in Table 3 seen on page 9.

- Line 222 – 223 “No consistent approach to discourage prescribing inappropriate medicines were observed” – this sentence would be considered more for the Discussion section. Consider describing existing high risk alert strategies observed at the study sites and discuss what kind of interventions should be considered e.g., alert sticker, clinical decision support.

As suggested, we have moved the indicated sentence from the results section to paragraph 5 of the discussion and expanded our discussion on this topic as seen on page 13. 

- Line 237 consider expanding more on the reason why alterations to medication formulation would contribute to high risk?

We have expanded the section on how different medication formulations may increase a patients risk of medication by including the following text as seen on line 250-254, “Within the Parkinson’s disease population, patients with a modified swallowing status or nil by mouth order were identified as ‘higher risk’, as they encountered significantly more medication errors and had an increased risk for longer length of hospital stay. If patients were unable to swallow their regular solid dose medications, they likely required an alternative medication formulation, this transition may have been responsible for the observed increased rate of medication errors for this subgroup.”

- Line 267 use lowercase ‘m’ for ‘medications’

We have replaced ‘Medicines’ with ‘medications’ in the highlighted sentence, this change can be viewed on line 284.

- Line 293 “… such practice need to become widespread” – this expression is a bit odd. Consider re-wording to something like “… such practices need to be promoted/shared/standardised within the healthcare setting” or something similar.

According to reviewer 1’s suggestion we have changed ‘become widespread’ to ‘promoted within the healthcare setting’, seen on line 314.

Reviewer #2: 

very good study I recommend publication after minor revision

I have few comments that might improve the manuscript:

Thank you. We appreciate the time and effort taken to improve our manuscript.

1-The methodology section can be presented better if have sub-headings on ethics, study design, study site, study population (inc/exc criteria), data collection, operational definitions, and statistics.

To improve clarity and flow of the methods section we have incorporated the sub-headings as suggested by reviewer 2, see methods section on pages 4-7.

2-The abbreviations in the tables must be defined in the footnotes

Footnotes with abbreviations have been added to Table 1 (page 8) and Table 4 (page 10) as these were the tables that included abbreviations within. 

3-Authors have used the term medication incidents, which is not very common in the pharmacy field. What if the author considers using the term "Medication errors or medication-related problems"?

We have replaced ‘medication incidents’ with ‘medication errors’ throughout the manuscript to align with the phrasing more commonly used in the pharmacy field.

4-Authors have evaluated the factors leading to medication incidents and in the methodology section, they have described those reasons of incidents were noted if they were presented in the medication chart. I have one confusion here if the pharmacist has visited the patient within 24 hours can be identified from the record, can the authors describe how other factors were identified? were they being routinely noted in the patients` records?

The pharmacist led medication history included the date the history was taken and is recorded on a separate document to the medication chart. The errors were identified in either notation on the patient’s medication chart or through the details provided on the pharmacist led medication history document. We have included the following sentence in the method section, line 117-119 to clarify this in our manuscript, “Records of patients seen by a pharmacist during their hospital stay included a pharmacist led medication history document in addition to their ‘National Inpatient Medication Chart’.”

We have also included how the other type of medication errors were derived from the patient’s charts, see line 134-141.

5-The conclusion section is broad, please be specific in conclusion with a separate heading.

We have separated out a conclusion and have included a sub-heading ‘conclusion’ at the end of the discussion section, line 333. 

6-This paper lacks recommendations for pharmacists, healthcare authorities, and physicians, as well as for future research.

We have included the following recommendations in the conclusion on page 14, “We recommend adjusting health system approaches to target these particular factors, and suggest working with people with Parkinson’s disease to utilize existing levers such as staff training, guideline development and clinical prioritization of people with Parkinson’s disease for pharmacist review when they first present to hospital. These strategies may represent achievable solutions to reduce preventable medication errors in this patient group”. Earlier in the discussion on line 284-286 we also state the following as a recommendation for hospital pharmacists “We believe these data indicate that people with Parkinson’s should be ranked as of ‘high importance’ when developing clinical prioritization tools for pharmacists”.

In the discussion we state a possible future direction in the following sentence “Repeating this study in the presence of electronic health records and wards with automated dispensing cabinets would improve the precision and accuracy of the data and reduce gaps in the information that we are able to pull from patient records”, seen on line 323-325. On line 331-332 we also state “From here future research could explore the root causes for common error types and interventions could be developed using corresponding evidence-based behavior change techniques”.

7-This study is accompanied by a few limitations as well as strengths, but the authors did not provide them at the end of the discussion section. For example, researcher bias is common in this study, the evaluation of factors all depends on the researcher`s ability to assess them. Strength is sample size from three large hospitals

As per reviewer 2’s suggestion we have adjusted the last paragraph of the discussion on page 13 to more specifically outline limitations and strengths of our study.

---

## [Editor Report · Decision Letter 1]

20 Apr 2022

Identifying rates and risk factors for medication errors during hospitalization in the Australian Parkinson’s disease population: A 3-year, multi-center study

PONE-D-21-35261R1

Dear Dr. Johnson,

We’re pleased to inform you that your manuscript has been judged scientifically suitable for publication and will be formally accepted for publication once it meets all outstanding technical requirements.

Kind regards,

Ismaeel Yunusa, PharmD, PhD

Academic Editor

PLOS ONE
---

## [Editor Report · Acceptance letter]

25 Apr 2022

PONE-D-21-35261R1 

Identifying rates and risk factors for medication errors during hospitalization in the Australian Parkinson’s disease population: A 3-year, multi-center study 

Dear Dr. Johnson:

I'm pleased to inform you that your manuscript has been deemed suitable for publication in PLOS ONE. Congratulations! Your manuscript is now with our production department. 

Kind regards, 

on behalf of

Dr. Ismaeel Yunusa 

Academic Editor

PLOS ONE